# Evaluation of Prescribing Indicators in a Paediatric Population Seen in an Outpatient Consultation at the Gaspard Kamara Health Centre in 2019 (Senegal)

**DOI:** 10.3390/pharmacy9020113

**Published:** 2021-06-17

**Authors:** Oumar Bassoum, Mouhamadou Faly Ba, Ndèye Marème Sougou, Djibril Fall, Adama Faye

**Affiliations:** 1Department of Public Health and Preventive Medicine, Faculty of Medicine, Pharmacy and Odontology, University Cheikh Anta Diop, 5005 Dakar, Senegal; ndeyemareme.sougou@ucad.edu.sn (N.M.S.); adama.faye@ucad.edu.sn (A.F.); 2Institute of Health and Development, University Cheikh Anta Diop, 16390 Dakar, Senegal; mouhamadoufaly.ba@ucad.edu.sn; 3Laboratory of Therapeutic and Organic Chemistry, Faculty of Medicine, Pharmacy and Odontology, University Cheikh Anta Diop, 5005 Dakar, Senegal; djibril.fall@ucad.edu.sn

**Keywords:** prescribing indicators, essential medicines, paediatrics, Senegal

## Abstract

*Introduction*: Data on drug use in paediatrics are scarce in Senegal. The objective of this study was to assess the prescribing indicators in a paediatric population seen in an outpatient consultation at a Health Centre in Dakar, Senegal. *Methods*: A retrospective and analytical study was conducted. The study population consisted of prescriptions for children aged 0 to 14 years who were seen in ambulatory consultation between 1 June and 30 November 2019. The sample size was 600 prescriptions. The systematic survey was then conducted. Five prescription indicators recommended by the World Health Organization were calculated. The R software was used for descriptive analysis, bivariate analysis and binomial logistic regression. *Results*: The average number of drugs per prescription was 2.56. The proportion of drugs prescribed under the International Nonproprietary Name (INN) was 18.9%, while the proportion of drugs on the National Essential Medicines List (NEML) was 41.3%. The proportions of prescriptions with at least one antibiotic and one injectable product were 41.5% and 1.3%, respectively. *Conclusions*: This study showed that prescribing habits were inadequate. Thus, it would be necessary to move towards continuing training of prescribers in the wise use of medicines.

## 1. Introduction

The rational use of medicines is to “prescribe the most appropriate product, obtained on time and at an affordable price for all, delivered correctly and administered in the appropriate dosage and for an appropriate period of time” [1]. Any use of drugs that does not meet this definition is considered irrational [2].

According to the World Health Organization (WHO), more than half of all medicines are prescribed, dispensed or sold inappropriately. These practices are more prevalent in developing countries where routine monitoring mechanisms for drug use are still at an embryonic stage [3].

The most common irrational practices are, among others, polypharmacy, inappropriate use of antimicrobials resulting in inadequate dosing or antibiotic-based treatment of nonmicrobial infections, the inappropriate and unsafe use of injectable products and the use of brand-name medicines resulting not only in low prescription of International Nonproprietary Name (INN) drugs but also in low prescribing on the basis of a National Essential Medicines List (NEML) [2].

There are many factors that contribute to the irrational use of drugs. These may be related to patients, prescribers, the work environment, the supply system, poor regulation of drug sales and lack of information on drugs [4].

The consequences can be very serious, resulting in increased mortality and morbidity, ineffective treatment, a breakdown in patient confidence in health systems, wasted drugs and resources and unnecessary spending [4]. In countries with limited resources, pharmaceuticals account for a high proportion of household spending [3]. These products can represent up to 70% of the total cost of healthcare [5].

The promotion of rational use of medicines is considered an essential intervention for the success of National Medicine Policies (NMPs). To this end, the WHO has defined several indicators to monitor the use of medicines. These include five prescription indicators: (i) average number of medicines per prescription, (ii) proportion of medicines prescribed under international nonproprietary names, (iii) proportion of medicines prescribed and included in the national essential medicines list, (iv) proportion of prescriptions containing at least one antibiotic and (v) proportion of prescriptions containing at least one injectable [6].

A systematic review of 43 studies conducted in the WHO African Region evaluating these indicators showed the poor quality of prescriptions in primary healthcare services [3]. However, prescriptions for children are under-represented in these studies. This age group is a significant population in developing countries and is more vulnerable to irrational prescriptions [7]. A retrospective analysis of 294 prescriptions prescribed in a paediatric hospital in the Sierra Leonean capital showed a high number of prescription drugs and excessive use of antibiotics and injectables [8]. In Tanzania, in a prospective study conducted in public health facilities in a health district, there has been a satisfactory buy-in by prescribers to the concept of essential drugs, which has resulted in a high use of generics and NEML [9]. In contrast, the audit of 600 prescriptions in a retrospective study conducted in the ambulatory care services of a paediatric hospital in Nigeria indicated low INN prescription and high use of penicillins [10]. However, to date, no such study has been conducted in Senegal. Nevertheless, the country has formulated an NMP [11] and developed various documents to promote the wise use of medicines [12,13]. Therefore, conducting studies on the quality of paediatric prescriptions is a necessity. It is in this context that an investigation was conducted at the Paediatric Department of the Gaspard Kamara Health Centre to assess prescribing practices and their compliance with WHO standards. The benefit of this study is that it measures the prescribing habits of health professionals [6] and, therefore, implements strategies to improve prescribing practices. The objective of this study was to evaluate the indicators of prescription in a paediatric population seen in outpatient consultation at the Gaspard Kamara Health Centre of the health district of Dakar Centre.

## 2. Materials and Methods

### 2.1. Study Site

The study was conducted at the Gaspard Kamara Health Centre. This centre is located in the capital of Senegal (Dakar) and belongs to the borough of Plateau and the borough of Fann-Point E-Amitié [14]. It is the reference centre of the Dakar Centre district and covers a population of 27,113 inhabitants including 11,312 children aged 0 to 14 years [14]. The Senegalese health system is organized according to a pyramid structure at three levels: central, intermediate and peripheral (MoH). The health districts are located at the peripheral level.

### 2.2. Type and Period of Study

It was a retrospective and descriptive study. The data collection took place from 11 to 20 December 2019.

### 2.3. Population

It consisted of prescriptions for children aged 0 to 14 years and seen in consultation during the period from 1 June to 30 November 2019 in the paediatric department of the Gaspard Kamara Health Centre.

### 2.4. Sampling

This study was aimed at prescriptions for children treated on an outpatient basis. A drug with several combinations of active ingredients was counted as one drug. Vaccines and parapharmacy products were not counted.

A sample size of 600 prescriptions was selected at a rate of 100 prescriptions per month. WHO recommends that prescription indicators be analysed on a sample of at least 600 prescriptions [15]. The selection was made using systematic sampling. For each month, the total number of prescriptions was established. The survey step was obtained by dividing this number by 100.

### 2.5. Data Collection

The collection tool was a modified WHO form [15] in which the data needed for the study were recorded. The information collected was sociodemographic characteristics (age and sex), clinical-anthropometric characteristics (weight, height, nutritional status), diagnosis and prescribing practices (the name of each drug prescribed, the number of drugs prescribed, the number of INN drugs prescribed, the number of drugs prescribed and listed on the NEML, the number of prescriptions containing at least one antibiotic and the number of prescriptions containing at least one injectable). Three sources of information were used. The first was the register of consultations of sick children aged 0–14 years. The investigator’s task was to fill in the items on the form based on the register. The second source of information was Senegal’s NEML published in 2019 [12]. It was to check whether or not the prescribed drugs were in there. The third source was the WHO International Classification of Diseases (ICD-10) document. This source was used to categorize diagnoses into broad classes of diseases [16].

### 2.6. Operational Definition of Variables

Index of Rational Drug Prescribing (IRDP): The IRDP consists of five indices derived from prescription indicators. Each indicator has an optimal index equal to 1. The closer the index is to 1, the better the indicator. This index was also used by Cole et al. in their study and calculated as in [8] to give numerator and denominator:The index of nonpolypharmacy is represented by the percentage of prescriptions with 3 drugs at most.The proportion of drugs prescribed by INN measured the generic name index.The essential medicine index was measured by the proportion of drugs prescribed and reported on the NELM.The index of rational antibiotic prescribing was defined as the ratio between the optimal level (30%) and the proportion of prescriptions containing at least one antibiotic.The index of rational injection prescribing was defined as the ratio between the optimal level (10%) and the proportion of prescriptions containing at least one injectable.

The IRDP, which has a maximum value of 5, can then be calculated by adding the indices.

### 2.7. Data Analysis

The data were analysed using R software version 3.6.6. The analysis consisted of describing sociodemographic characteristics, anthropometric characteristics, drugs prescribed, clinical characteristics, rational prescription and the IRDP. For this purpose, quantitative variables were expressed as a mean and its standard deviation, while qualitative variables were described as absolute and relative frequencies.

## 3. Results

### 3.1. Sociodemographic Characteristics

In all, 53.0% (318/600) of the prescriptions in study were prescribed to male children. The proportion of prescriptions for children aged 0 to 59 months was 77.7% (466/600).

### 3.2. Anthropometric Characteristics and Drugs Prescribed

The proportion of prescriptions for children with high body temperature was 59.5%. The average weight and average height were 14.2 ± 10.4 kg and 76.0 ± 17.9 cm, respectively. The proportion of prescriptions for children aged 1–59 months with acute malnutrition was 3.7% (Table 1). The total number of drugs prescribed was 1533. The proportion of prescriptions containing two drugs was 36.3%. The proportion of beta-lactam antibiotics prescribed in prescriptions containing at least one antibiotic (*n* = 249) was 70.0%. The 15 most frequently prescribed drugs are listed in Table 1. They represented about 63.3% of all drugs. Paracetamol was the most prescribed drug during the study period.

### 3.3. Clinical Characteristics

The proportion of prescriptions for children with respiratory diseases was 42.7%. Diseases of the digestive system accounted for 14.0% of the study (Table 2).

### 3.4. Prescribing Practices

The average number of drugs per prescription was 2.56 ± 1.35. The ranges were 1 and 10. The median number of drugs was 2. Of the values describing prescription indicators, only those containing at least one injectable drug (1.3%) met WHO standards (Table 3).

### 3.5. IRDP

The IRDP, used as an indicator of rational drug use, was 3.11. The INN prescription had the lowest index of 0.19 (Table 4).

## 4. Discussion

This study showed that prescription indicators, except for the use of injectables, do not comply with WHO standards. This irrational use of drugs was manifested by polypharmacy, noncompliance with INN prescription, low adherence to NEML and excessive use of antibiotics. The low value of the IRDP is evidence of this irrational use of drugs.

In this study, the average number of drugs per prescription was 2.56. Similar results were found in Oman (2.3; 2.88) [4,17], Pakistan (2.34) [18], Maharashtra (2.35) [19], India (2.29) [20] and Italy (2.9) [21]. Higher values were found in Sierra Leone (3.77) [8], Nepal (4.5) [22] and Nigeria (3.5; 3.8) [7]. A number greater than two is indicative of a trend towards polypharmacy. There is a risk of drug interactions and adverse drug reactions (ADRs) [23] that is directly related to the number and potency of drugs administered [24]. It is also associated with increased costs and noncompliance with treatment [25]. Polypharmacy may result from financial incentives to prescribers, pressure from medical delegates and inadequate training of prescribers in the rational use of medicines [7].

The proportion of INN drugs prescribed was 18.9%. This result is similar to data found in Switzerland (24.7%; 20.9%) [26]. The influence of medical delegates on prescribing habits through benefits and gifts offered to prescribers on the one hand [27,28] and the low level of knowledge of prescribers on the concept of generic medicine on the other hand [29] could be the cause of this situation. At the same time, in the public sector, drug disruptions are recurrent in Senegal [30]. Thus, prescribers are forced to use brand-name drugs (originator drugs). However, prescribing INN drugs has certain benefits. A survey in Belgium showed that it reduced government and patient spending on pharmaceuticals and that patients even perceive INN prescribing as a way to help rationalize prescribing [31]. It also reduces the risk of confusion during dispensing [32] and facilitates communication between health professionals [8].

The proportion of drugs prescribed and reported on the NEML was 41.3%. This shows a low adherence of prescribers to Senegal’s NEML. The value of this indicator is similar to that found in Oman (45.1%) [17] but lower than those found in Nigeria (85.6%; 86.8%) [7], Sierra Leone (70.6%) [8] and Pakistan (87.1%) [18]. The low dissemination of the NEML and the influence of medical delegates have been shown to be factors that may explain this situation [3]. Further studies should be conducted to assess the availability of the NEML to prescribers.

In addition, the proportion of prescriptions containing at least one antibiotic was 41.50%. This result is similar to that found in Malaysia (43.5%) [33]. Significantly higher results are found in Nigeria (77.1%; 72.3%) [7], India (73.2%) [20] and Sierra Leone (74.8%) [8]. In contrast, WHO-compliant values were noted in Oman (15.9%) [17] and Pakistan (24.3%) [18]. Irrational use of antibiotics has been shown to be a factor that accelerates bacterial resistance. In West Africa, difficult socioeconomic conditions have exacerbated this phenomenon [34].

The proportion of prescriptions containing at least one injectable was 1.3%. Similar results were found in India (0.006%) [20]. Above-standard proportions were noted in studies conducted in Nigeria (10.2%; 25.3%) [7], Oman (15.2%) [17], Pakistan (19.9%) [18] and Sierra Leone (21.1%) [8]. It has been shown that the increased use of injectable products contributes to increased hospital waste when waste management systems are not fully effective, causes unnecessary pain to the patient and promotes the occurrence of blood-borne infections. Therefore, the rational use of injectables highlighted in this study is encouraging as it helps to avoid the occurrence of these events.

This study has some limitations. The study did not use recent tools such as POPI (Paediatrics: Omission of Prescriptions and Inappropriate Prescriptions) or PIPc (Potentially Inappropriate Prescribing in Children) [35,36]. The study did not take into account seasonal variations that could influence the epidemiological profile and prescribing practices. However, the six-month decline appears to mitigate the effects of this limitation. The study also did not take into account the prescribing patterns and the rational prescription for each diagnosis during this period. In addition, the study was carried out in a single urban structure. Thus, the results are not generalized to the whole of the country. However, it can be said that the districts are organized in the same way and the providers trained in the same schools with some degree of homogeneity.

## 5. Conclusions

This study revealed that all prescription indicators in the Gaspard Kamara Health Centre deviated from WHO standards except for the number of prescriptions containing at least one injectable product. Thus, it would be necessary to promote the rational use of medicines through a programme of continuing education for prescribers and regular evaluation of prescriptions. It would also be necessary to implement antibiotic stewardship programmes to improve antibiotic prescribing and limit the development of antimicrobial-resistant organisms. In addition, qualitative research may be needed to understand the reasons that led prescribers to engage in irrational drug use.

## Figures and Tables

**Table 1 pharmacy-09-00113-t001:** Anthropometric characteristics and drugs prescribed of children seen in consultation at Gaspard Kamara HC.

Anthropometric Characteristics and Drugs Prescribed	Number (*n*)	Percentage (%)
Nutritional status (*N* = 455)		
Severe acute malnutrition	4	0.9
Moderate acute malnutrition	13	2.9
Alert	23	5.0
Normal	238	52.3
Overweight	177	38.9
The 15 drugs most frequently prescribed (*N* = 1533)		
Paracetamol (acetaminophen)	161	10.5
Paracetamol + Chlorpheniramine + Phenylephrine	147	9.6
Amoxicillin	99	6.5
Normal saline (nose drops)	92	6.0
Paracetamol + Ibuprofen	73	4.8
Guaifenesin	55	3.6
Mequitazine	48	3.1
ORS	45	2.9
Zinc	45	2.9
Salbutamol	44	2.9
Chlorhexidine	38	2.5
Alpha-amylase	36	2.3
Amoxicillin—clavulanic acid	32	2.1
Cefixime	28	1.8
Azithromycin	27	1.8

**Table 2 pharmacy-09-00113-t002:** Clinical characteristics of children seen in consultation at Gaspard Kamara HC (*N* = 599).

Clinical Characteristics	*n*	%
Diseases of the respiratory system		
Upper respiratory tract	245	40.9
Lower respiratory tract	11	1.8
Diseases of the digestive system		
Diarrhoea	54	9.0
Colic	20	3.3
Aphthous ulceration	10	1.7
Symptoms not classified elsewhere		
Fever	44	7.3
Abdominal pain	29	4.8
Cough alone	9	1.5
Skin and tissue diseases under skin		
Allergy	27	4.5
Dermatosis	21	3.5
Erythema	19	3.1
Traumatic injuries		
Injury	35	5.8
Trauma	6	1.0
Infectious and parasitic diseases		
Infectious syndrome	22	3.7
Varicella	7	1.1
Diseases of the eye		
Conjunctivitis	17	2.8
Eye secretion	1	0.2
Diseases of the ear		
ENT illness	13	2.2
Suspicion of disease	7	1.2
Diseases of the genitourinary tract		
Urinary tract infection	2	0.3
Unknown diagnosis	2	0.3

**Table 3 pharmacy-09-00113-t003:** External consultation prescription indicators seen at Gaspard Kamara HC (*N* = 600).

Indicators	Values	95% CI	WHO Standards
Average number of drugs per prescription	2.56 ± 1.35	[2.45–2.67]	2
Proportion of INN drugs prescribed	18.9%	[15.9–22.1%]	100%
Proportion of drugs prescribed from the NELM	41.3%	[37.1–44.9%]	100%
Percentage of prescriptions with an antibiotic prescribed	41.5%	[37.6–45.5%]	<30%
Percentage of encounters with an injection prescribed	1.3%	[0.7–2.6%]	<10%

**Table 4 pharmacy-09-00113-t004:** IRDP at the paediatric service of the Gaspard Kamara Health Centre.

Index of Nonpolypharmacy	Essential Medicine Index	Generic Name Index	Index of Rational Antibiotic Prescribing	Index of Rational Injection Prescribing	IRDP
0.79	0.41	0.19	0.72	1	3.11

## Data Availability

Data are available upon request to the authors.

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
