# Peer review of "Evaluation of Prescribing Indicators in a Paediatric Population Seen in an Outpatient Consultation at the Gaspard Kamara Health Centre in 2019 (Senegal)"

_pharmacy, 2021, doi:10.3390/pharmacy9020113_

Round 1

Reviewer 1 Report

The manuscript consists of an original and interesting retrospective, descriptive and analytical study that aimed to evaluate the prescription indicators in a paediatric population seen in an outpatient consultation at a health centre in Dakar, Senegal, as well as their compliance with WHO standards.

The importance of the study stems from the fact that prescriptions for children are underrepresented in the studies currently available for the WHO African Region, despite that this age group is a significant portion of the population in developing countries and is more vulnerable to irrational prescriptions. In addition, no such study has been conducted in Senegal.

The study showed that prescribing habits were inadequate, not meeting WHO standards, except for the use of injectables. The low value of the Index of Rational Drug Prescribing (IRDP) is evidence of the irrational use of drugs. The reasons underlying the inadequacy of prescription habits are discussed, and measures leading to the promotion of the rational use of medicines are presented.

The study was designed and carried out properly and the conclusions are well supported by the data obtained.

As a general recommendation, the manuscript should be carefully revised, in order to improve the English language and style. For example, some uniformity in writing is required (e.g., paediatric vs pediatric; “Health Centre” vs “health center”).

Major remarks:

Line 42

Considering that polypharmacy corresponds to the use of more than 3 drugs does not seem to be supported by the corresponding reference [2]. Better clarification of the concept and the inclusion of a bibliographic reference that supports the statement is recommended.

In fact, there is no consensus on the number of drugs to be considered polypharmacy. Most papers consider that polypharmacy corresponds to the simultaneous use of five or more medications (e.g., Masnoon, N., Shakib, S., Kalisch-Ellett, L., & Caughey, G. E. (2017). What is polypharmacy? A systematic review of definitions. BMC geriatrics, 17(1), 230. https://doi.org/10.1186/s12877-017-0621-2)

Line 71

There is a typo in “… district, There has been…”.

“there” should be written in lowercase.

Line 90

There is a typo: Fann-Point E-AMITIÉ should be replaced by Fann-Point E-Amitié

Lines 128 to 130

The authors state that in the present study, “polypharmacy is defined when the number of drugs prescribed is greater than three”. The inclusion of a bibliographic reference supporting the statement is recommended.

Line 165

There is a typo in the sentence “The proportion of children with normal body temperature was 59.52%”.

“…normal…” should be replaced by “high”

Table 1. (pages 4 and 5)

N corresponding to “Body Temperature” is 583. Nevertheless, the sum of the parts – “Bass”, “Normal” and “High” – is 581.

The difference should be clarified.

Line 185

There is a typo in “The Only body temperature …”.

“only” should be written in lowercase.

Lines 193 to 195

“(…) children who were prescribed 3 and at least 4 medicines were 2.73 and 19.89 times more likely (…).”

It is recommended that the sentence should be rewritten as follows (inserting the word "respectively"): “(…) children who were prescribed 3 and at least 4 medicines were respectively 2.73 and 19.89 times more likely (…).”

Line 205

“Other higher were found in Sierra Leone (…)”

It is recommended that the sentence should be rewritten as follows: “Higher values were found in Sierra Leone (…).”

Author Response

Good morning,

Sincerely

Reviewer 2 Report

This well-written manuscript describes a retrospective review designed to determine the appropriateness of medication prescribing in an outpatient pediatric population in Senegal. The design is sound and evidence-based, and the results are presented appropriately. I agree with your discussion points and conclusions. I have two suggestions that you may want to address in your discussion.

  1. Firstly, your data collection period is quite short (June through November 2019). You have touched on this a bit under study limitations, but it may merit more discussion. Did you look at any possible prescribing patterns by month? My understanding is that in Senegal influenza cases peak during the rainy season (between July and October). Roughly 40% of your subjects had a high temperature, and 43% respiratory disease. Do you know what percentage had a diagnosis of influenza, and how would this be expected to affect prescribing? Do you know what the prescribing patterns look like during the remainder of the year?
  2. Secondly, you may want to discuss the need for and potential impact of an antibiotic stewardship program on antibiotic prescribing. Numerous publications have described the impact of these programs in both the inpatient and outpatient setting.

Author Response

Good morning,

Sincerely.

Reviewer 3 Report

This is an important topic. However, the authors need to expand their results considerably. Diagnoses need to be more specific, eg Diseases of the respiratory system will include pneumonia where antibiotics are indicated and bronchiolitis where they are not indicated. List clinical diagnoses.

Similarly the actual medicines prescribed need to be listed.

Delete statistical analyses. Use descriptive stats only.

The literature regarding rational prescribing in paediatrics is inadequately reviewed. The tools used to determine rational prescribing are old. See     Choonara I. Evaluation of rational prescribing in paediatrics. BMJ Paediatr Open 2021;5:e001045. doi: 10.1136/bmjpo-2021-001045 for more recent tools.

Use one decimal point only for %

Author Response

Good morning,

Sincerely.

Round 2

Reviewer 2 Report

Thank you for addressing my comments.

Author Response

Please see the attachment. Thank You.

Sincerely.

Reviewer 3 Report

Disappointing to see that the authors have failed to respond to all the previous points. The paper in its present format does not help readers understand the problems of irrational prescribing in children

Author Response

(The authors gave the same response as above.)

Round 3

Reviewer 3 Report

The paper has improved, but there are still problems. The diagnoses in Table 1 need to be separated, eg resp disease needs to be lower respiratory tract infections (pneumonia); asthma; tonsillitis;upper respiratory tract infection. The numbers and the treatment given for each condition need be listed separately. Which children received the antibiotics? (ie was it rationalprescribing or not?).The section on temp is irrelevant.

Author Response

Hello.

Sincerely.
